# Foliage of Tropical Trees and Shrubs and Their Secondary Metabolites Modify In Vitro Ruminal Fermentation, Methane and Gas Production without a Tight Correlation with the Microbiota

**DOI:** 10.3390/ani12192628

**Published:** 2022-09-30

**Authors:** Yesenia Ángeles-Mayorga, Elmi Roseida Cen-Cen, María Magdalena Crosby-Galván, Jacinto Efrén Ramírez-Bribiesca, Bernardino Candelaria-Martínez, Alfredo Sánchez-Villarreal, Mónica Ramírez-Mella

**Affiliations:** 1Recursos Genéticos y Productividad-Ganadería, Colegio de Postgraduados Campus Montecillo, Texcoco 56230, Estado de México, Mexico; 2Bioprospección y Sustentabilidad Agrícola en el Trópico, Colegio de Postgraduados Campus Campeche, Sihochac 24450, Campeche, Mexico; 3Agroecosistemas Sostenibles, Tecnológico Nacional de México Campus Instituto Tecnológico de Chiná, Chiná 24520, Campeche, Mexico; 4Bioprospección y Sustentabilidad Agrícola en el Trópico, CONACYT-Colegio de Postgraduados Campus Campeche, Sihochac 24450, Campeche, Mexico

**Keywords:** ruminant, condensed tannins, saponins, greenhouse gases, digital droplet PCR, archaea, protozoa

## Abstract

**Simple Summary:**

Ruminants produce methane, a potent greenhouse gas, as a byproduct of microbial activity during ruminal fermentation. To lessen greenhouse gases emissions, it is necessary to evaluate methane-reducing feeding alternatives for ruminants, such as the use of tropical trees and shrubs. The secondary metabolites produced by these plants have been shown to reduce ruminal methane emissions. Here, we evaluated the effect of the foliage of eleven tropical trees and shrubs edible for cattle in ruminal fermentation, methane production and microbiota in an in vitro assay. The content of condensed tannins and saponins in the foliage was variable between the species of plants. We observed a reduction in methane production and changes in microbial populations depending on the species of tree or shrub. Additionally, condensed tannins reduced methane production. The inclusion of the foliage of tropical trees and shrubs in the diet of ruminants may be a step towards diminishing greenhouses gases emissions, and therefore reducing the contribution to global warming from cattle production.

**Abstract:**

Ruminants, mainly cattle, contribute to greenhouse gases (GHG) emissions as methane (CH_4_) is produced by ruminal fermentation. Hence, various anti-methanogenic feed strategies have been studied, including the use of plants with secondary metabolites. This study evaluated in vitro ruminal fermentation metrics, microbial composition by digital droplet PCR (ddPCR) and the CH_4_ production of the foliage of several tropical trees and shrubs: *Leucaena leucocephala*, *Moringa oleifera*, *Albizia lebbeck*, *Enterolobium cyclocarpum*, *Piscidia piscipula*, *Brosimum alicastrum*, *Lysiloma latisiliquum*, *Guazuma ulmifolia*, *Cnidoscolus aconitifolius*, *Gliricidia sepium* and *Bursera simaruba*, using *Cynodon plectostachyus* grass as control. The results showed a wide variation in the chemical composition of the foliage, as well as in the ruminal microbiota. The crude protein (CP) content ranged from 11 to 25%, whereas the content of condensed tannins (CT) and saponins (S) was from 0.02 to 7%, and 3.2 to 6.6%, respectively. The greatest dry matter degradability (DMD) after 72 h was 69% and the least 35%, the latter coinciding with the least gas production (GP). A negative correlation was found between the CT and CH_4_ production, also between protozoa and fungi with the SGMT group of archaea. We concluded that the foliage of some tropical trees and shrubs has a high nutritional value and the potential to decrease CH_4_ production due to its CT content.

## 1. Introduction

By 2050, the world population will reach 9.6 billion [1] and the demand for milk and meat will increase by 73% and 58%, respectively [2]. Consequently, the farmland necessary for forage and grains to feed livestock will increase, as well as the amount of greenhouse gases (GHG) emitted into the environment [3]. Livestock contributes to anthropogenic GHG emissions with approximately 14.5% of the total world production [2]. Worldwide, Latin America and the Caribbean regions have the highest level of GHG emissions, producing 1.9 gigatons of CO_2_-eq, originating mainly from the production of beef cattle [4]. This may be due to the fact that in tropical regions, such as those found in Latin America and the Caribbean, ruminants are commonly fed with low-quality forages, particularly in the dry season. During this season, forage crude protein (CP) decreases and neutral detergent fiber (NDF) increases. Consequently, the fermentation of DM is affected, retention time of digesta increases and VFA absorption decreases in the rumen, thus limiting the cattle productive performance. Therefore, the CH_4_ production per unit of milk or meat is increased [5].

Ruminants produce approximately 80 million tons of methane (CH_4_) annually, representing 28% of anthropogenic emissions [6]. Cattle contribute approximately 65% of these emissions which is relevant [2] as CH_4_ has a higher global warming potential (a relative measure of the amount of heat held in the atmosphere) of 28 times more than that of CO_2_ [7]. Most of the CH_4_ produced by ruminants is generated by archaea through the reduction of CO_2_ with H_2_, both byproducts of bacterial, anaerobic fungi and protozoa fermentation processes [8]. It is estimated that an adult bovine generates from 250 to 500 L of CH_4_ per day, representing a loss of 2 to 12% of the energy consumed [9].

Various strategies have been described to reduce ruminal methanogenesis, including a modification of the composition of the diet to increase the passage rate or a higher propionate/acetate rate, the use of antibiotics [6], anti-methanogenic vaccines [10], methyl-coenzyme M reductase (MCR) inhibitors [10,11], defaunation [12] and food additives such as fats [13], nitrates [14] or secondary metabolites, such as condensed tannins (CT) and saponins (S), found in various forage plants [8,15]. In this regard, some secondary metabolites have been shown to have an antimicrobial activity and reduce the availability of H_2_, which is used by methanogenic archaea to produce CH_4_ [16].

Several studies have shown that the foliage of several tropical trees and shrubs decrease the production of ruminal CH_4_ [17,18,19,20,21], an effect attributed to the content of various secondary metabolites [21]. Furthermore, plant CT and S modify, to some extent, microbial populations distribution in the rumen [22]. Hence, this study aimed to evaluate the effect of the foliage of tropical trees and shrubs, and their secondary metabolites, on in vitro ruminal degradation, short-chain fatty acids (SCFA), CH_4_ and gas production (GP) and microbial composition.

## 2. Materials and Methods

### 2.1. Foliage Collection and Treatments

The foliage of trees and shrubs was collected in June and July of 2016 at Colegio de Postgraduados Campus Campeche and nearby towns in Champotón, Campeche, Mexico, simulating the browsing of an adult bovine at a maximum height of two meters. Five kg of fresh and young leaves from several plants per species were collected [17]. The foliage of each of the following eleven species of trees and shrubs were used as treatments: *Leucaena leucocephala*, *Moringa oleifera*, *Albizia lebbeck*, *Enterolobium cyclocarpum*, *Piscidia piscipula*, *Brosimum alicastrum*, *Lysiloma latisiliquum*, *Guazuma ulmifolia*, *Cnidoscolus aconitifolius*, *Gliricidia sepium* and *Bursera simaruba* which were all identified according to the CONABIO guidelines [23]. *Cynodon plectostachyus*, a monocotyledonous plant commonly used as forage for livestock in tropical regions [24], was used as control. The samples were dried at 50 °C in a forced air oven for 24–72 h until at a constant weight. Subsequently, they were grounded in a Thomas-Wiley Mill with a 1 mm mesh.

### 2.2. In Vitro Fermentation

This study was approved by the Comité Académico del Programa de Ganadería of Colegio de Postgraduados Campus Montecillo (GAN-16/264). The ruminal fluid was collected via a ruminal cannula from three Holstein steers, weighing approximately 500 kg, fed with a daily ration of 14 kg (DM basis), divided into two feeding times at 07:00 and 19:00 h. The basal diet consisted of 15% ground corn, 15% soybean meal, 61.5% corn stover, 6% molasses, 0.5% salt, 1.4% dibasic calcium phosphate and 0.6% calcium carbonate (DM basis), formulated according to the requirements suggested by the NRC [25]; water was offered ad libitum. The ruminal fluid was collected in the morning one hour before feeding. The ruminal fluid was filtered through four layers of cheesecloth, placed in a thermos flask preheated to 39 °C and was immediately transported to the laboratory, which is located ≈150 m from the pens. Once there, the rumen liquid was used for the inoculum preparation taking care to maintain a temperature of 39 °C and a constant flow of CO_2_.

The preparation of the inoculum was carried out according to Menke and Steingass [26]. Before the inoculum preparation, 0.5 g of the sample was placed in a 120 mL glass vial. Subsequently, 50 mL of inoculum was placed in each vial, under a constant flow of CO_2_, covered with a rubber stopper and an aluminum ring, and placed in a water bath at 39 °C with manual shaking occurring every 2 h. Additionally, three vials were incubated only with inoculum as blanks. Three incubations (runs), each with triplicate flasks per treatment, were carried out.

### 2.3. Dry Matter Degradability

The dry matter degradability (DMD: mg/g DM) [26] was determined at 6, 12, 24, 48 and 72 h. Three vials were incubated for each of the aforementioned times. The fermentation and activity of the ruminal microorganisms were stopped by removing the vials from the water bath and placing them in a refrigerator at 4 °C for 2 h. Afterwards, the vials content was filtered using a Whatman filter paper with a pore size of 22 µm. The filtered sample was dried at 70 °C for 24 h and weighed in order to calculate the DMD.

### 2.4. In Vitro Gas Production

The GP was recorded at 6, 12, 24, 48 and 72 h of fermentation, by the displacement of water according to the Fedorah and Hrudey methodology [27]. Three vials were incubated for each of the aforementioned times. The rubber stopper of the vial was punctured with a 1.2 × 40 mm needle and the displaced milliliters of water in the glass burette, equivalent to the GP, were recorded.

### 2.5. Determination of CH_4_ and CO_2_

The determination of CH_4_ and CO_2_ was carried out at 24 h of in vitro fermentation, thus taking into account only the gas produced between 12 and 24 h. With a 5 mL plastic syringe, 3 mL of gas were collected and added to a 60 mL vial filled with saturated saline solution (without leaving air space) capped with a rubber stopper and sealed with a metal ring [28]. The gas samples stored in the vials with saturated saline solution were kept in the dark at 4 °C until their analysis. CH_4_ and CO_2_ were analyzed in a gas chromatograph (Perkin Elmer model Clarus 500, PerkinElmer, Inc., Waltham, MA, USA) equipped with a Porapak capillary column with the following conditions: the oven at 80 °C, column at 170 °C and thermal conductivity detector at 130 °C. The standard used was a mixture of CH_4_ (25%) and CO_2_ (75%). Helium (22.3 mL/min) was the carrier gas. The retention times were 0.62 and 0.92 min for CH_4_ and CO_2_, respectively [29].

### 2.6. Production of Short-Chain Fatty Acids

At 72 h of fermentation, the content of each vial was placed in a 50 mL plastic tube and centrifuged at 11,319× *g* for 10 min in a benchtop centrifuge. Afterwards, 4 mL of the supernatant was collected, placed in 15 mL plastic tubes containing 1 mL of 25% metaphosphoric acid, and mixed by stirring. Finally, 1 mL was collected in a 1.5 mL microtube and centrifuged at 12,281× *g* for 10 min, then the supernatant was collected and stored at 4 °C. The determination of the SCFA was performed by gas chromatography (Hewlett Packard model 6890, Hewlett Packard, Palo Alto, CA, USA) [30] with the following conditions: the oven temperature at 140 °C, detector temperature at 250 °C with air (330 mL/min) and hydrogen (33 mL/min) flux and injector temperature at 240 °C using nitrogen (120 mL/min) as the carrier gas. The total retention time was 8.6 min.

### 2.7. Chemical Analysis

The foliage of the trees and shrubs as well as the grass were analyzed in triplicate to determine the dry matter (DM), ash, crude protein (CP) and ether extract (EE) according to the methods of the AOAC [31]. The neutral detergent fiber (NDF) and acid detergent fiber (ADF) were determined according to Van Soest et al. [32]. The CT and S content was performed according to the ISO 9648 [33] and Lee et al. [34] methodologies, respectively.

### 2.8. DNA Extraction

One vial from the in vitro fermentation incubated for 24 h from each treatment was used for the DNA extraction from the ruminal microbiota. The vial was lightly shaken before taking 1.5 mL of the content with a micropipette and placing it into a microtube, which was centrifuged at 200× *g* for 2 min, and the supernatant was recovered in a new microtube. Then, the microtube was centrifuged at 12,000× *g* for 5 min and the supernatant was discarded. The procedure was repeated and the tube was immediately frozen in liquid nitrogen.

For the DNA extraction, 1 mL of lysis solution (500 mM NaCl, 50 mM Tris-HCl, pH 8.0, 50 mM EDTA and 4% SDS) [35] was added to each pellet from the three incubation runs and resuspended with a vortex. Equal amounts (650 μL) of each sample were transferred to a 2.0 mL screw-cap tube, containing 0.1 g each of 0.1 mm and 0.5 mm silica beads, and it was homogenized at top speed for 3 min in a BeadBug (Benchmarck Scientific, Sayreville, NJ, USA). The microtubes were centrifuged at 16,000× *g* for 5 min and the supernatant was transferred to a fresh 2.0 mL microtube and an equal volume of phenol-chloroform-isoamyl alcohol (25:24:1) (Sigma, Saint Louis, MO, USA) was added, mixed with a vortex and again centrifuged at 16,000× *g* for 5 min. The upper phase was recovered, followed by a 0.2 mL chloroform extraction and centrifugation under the same conditions. The aqueous phase was recovered and precipitated with 0.2 vol. of ammonium acetate (0.3 M) and 1 vol. of ethanol at 4 °C for 30 min. The microtubes were centrifuged at 16,000× *g* for 5 min, the supernatant was discarded and the pellet was washed twice with 75% ethanol. Finally, the DNA was resuspended in nuclease-free water and cleaned with a OneStep PCR inhibitor removal kit (Zymo Research, Irvine, CA, USA) following the manufacturer’s instructions. The DNA quality and quantity was assessed by 1.2% gel electrophoresis and spectrophotometry with a Nanodrop (Thermo Fisher Scientific, Waltham, MA, USA).

### 2.9. Quantification of Ruminal Microbiota by Digital Droplet PCR (ddPCR)

The four main ruminal microbial populations were quantified by targeting the small subunit of the ribosomal RNA (SSU rRNA), 16S subunit for bacteria and archaea and the 18S for fungi and protozoa. Furthermore, to gain an insight into the subgroups of archaea, specific primers to target *Methanobrevibacter* sp. and the gene *methyl co-enzyme M reductase alpha subunit* (*mcrA*) (see Appendix A), which catalyzes the last step of methane production, were employed. Prior to ddPCR, the specificity of the primers were analyzed both in silico by BLAST and in the wet-lab, and the optimal Tm was determined by a gradient PCR assays.

For the DNA quantification we used the Bio-Rad QX200 system (Bio-Rad, Hercules, CA, USA) for ddPCR with an EvaGreen Supermix (Bio-Rad, Hercules, CA, USA) following the manufacturer’s recommendations. Briefly, quintuple 20 µL reactions for each sample were setup with 10 ng of DNA (except 16S SGMT which used 20 ng) and 250 nM of each primer (Appendix A). The ddPCR reactions were loaded in the DG8 cartridge and mixed with oil with the QX200 droplet generator. The top 40 µL of the mixture containing the droplets were gently transferred to a Twin-Tec 96 well plate (Eppendorf, Hamburg, Germany), and sealed with a pierceable foil in a PX1 (Bio-Rad). Then, the plate was transferred for thermal cycling in a T100 (Bio-Rad) with the following conditions: 95 °C/5 min, 40 cycles of 95 °C/30 s, X °C/60 s (where X is variable for each of the primers from Appendix A) with a ramp rate of 2.5 °C/s, then followed by 4 °C/5 min, 90 °C/5 min and 4 °C/infinite. After remaining overnight at 4 °C, the plate was brought to room temperature before loading it in the QX200 droplet reader.

Once the run was completed, the quantification analysis was performed with Bio-Rad´s proprietary software Quantasoft Analysis Pro V1.0 (Bio-Rad, Hercules, CA, USA). The data were filtered according to the software manual, thus discarding those samples that failed to produce a minimum of 10,000 droplets. A single threshold was set for the entire plate using the no template control as a guideline. After verifying manually both of the above for each sample in each plate, the software yielded the quantity of target DNA per sample as copies/μL in a 20 μL PCR reaction.

### 2.10. Calculations

The GP kinetics were calculated according to the model described by France et al. [36]:y = b × [1 − e^−c(t−L)^](1)
where y = the volume of gas at time t (h); b = the asymptotic gas production (Vmax; mL/g DM); c = fermentation fractional rate (Rate; /h); and L = lag phase (Lag; h).

The gas produced between 12 and 24 h of fermentation (GP_12–24_) was calculated as:GP_12–24_ = GP_24_ − GP_12_(2)
where GP_12–24_ = gas produced between 12 and 24 h of fermentation (mL/g DM); GP_24_ = accumulated gas production at 24 h (mL/g DM); and GP_12_ = accumulated gas production at 12 h (mL/g DM).

The CH_4_ and CO_2_ produced between 12 and 24 h of fermentation (CH_4_ mL or CO_2_ mL) were calculated as follows:CH_4_ mL or CO_2_ mL= [(GP_12–24_)(CH_4_% or CO_2_%)]/100(3)
where CH_4_ mL or CO_2_ mL = CH_4_ or CO_2_ produced between 12 and 24 h of fermentation (mL/g DM); GP_12–24_ = gas produced between 12 and 24 h of fermentation (mL/g DM); and CH_4_% or CO_2_% = percentage of CH_4_ or CO_2_ in GP_12–24_.

### 2.11. Experimental Design and Statistical Analysis

The experimental design was in completely randomized blocks. The results were analyzed with the GLM procedure of the statistical program SAS V. 9.0 (SAS Institute Inc., Cary, NC, USA) [37] and the comparison of means by the Tukey test was performed (*p* < 0.05). The model used was as follows:Yij = µ + ᵟi + ßj + Eij(4)
where Yij = the observable random variable of the ith treatment (foliage) with the jth repetition; µ = general mean; ᵟi = effect of the i-th treatment (foliage); ßj = effect of the j-th block (run); and Eij = experimental error.

The determination of the amount in DNA copies/µL of the targeted microbial groups is performed automatically by the QX200 system and Bio-Rad’s proprietary software (see Bio-Rad’s Digital Droplet Applications Guide bulletin 6407 which can be found on the manufacturer website). Briefly, the digital PCR QX200 system generates droplets, thus partitioning the sample with a random distribution. Each partition (droplets) may or may not contain one or more copies of the targeted DNA (template). After the PCR, the QX200 system separates and reads the droplets, individually detecting the positive from the negative, and consequently estimates the DNA concentration as copies/µL by modelling a Poisson distribution with 95% confidence intervals.

To estimate the degree of association between the CT and S content of the foliage, in vitro ruminal characteristics and microbial composition, a Spearman correlation analysis was performed under default parameters with a *p* = 0.05 with XLSTAT Basic software V2022.1 (Addinsoft, New York, NY, USA) [38]. Additionally, in order to gain insight into the relationship between all of the measured variables, a principal component analysis (PCA) under default parameters was executed, considering all 19 factors with XLSTAT software [38].

## 3. Results

### 3.1. Chemical Analysis

The chemical analysis showed variations between the different species of trees and shrubs and the grass (Table 1). As expected, *C. plectostachyus* had the least nutritional value, displaying the lowest CP content, the greatest NDF content, it ranked fourth with the highest content of ADF and the second with the lowest content of CT and ME. Both *L. leucocephala* and *M. oleifera* exhibited the greatest CP content, with more than 20% of DM, while *B. simaruba* was slightly higher than *C. plectostachyus*. On the other hand, *M. oleifera* and *C. aconitifolius* had the least contents of NDF and ADF (16 and 21% of NDF and 10 and 18% of ADF, respectively), whereas the rest of the trees and shrubs had values between 43 and 67% of NDF and between 27 and 45% ADF. It should be noted that *P. piscipula*, *B. simaruba* and *E. cyclocarpum* had a greater ADF content (43–45%) than *C. plectostachyus* (39%). The ash content ranged from 4.3% (*L. leucocephala*) to almost 12% (*B. simaruba*). The greatest CT content was found with *G. ulmifolia* (7%), and the least with *M. oleifera* (0.02%).

### 3.2. Dry Matter Degradability

The DMD of the foliage of the trees and shrubs as well as in the grass displayed significant differences (*p* < 0.05). In Table 2, it can be observed that *B. alicastrum* had a DMD of almost 70%, *C. aconitifolius*, *M. oleifera*, *A. lebbeck* and *C. plectostachyus* had a DMD above 50%, whereas *G. ulmifolia*, *P. piscipula*, *L. leucocephala*, *E. cyclocarpum* and *L. latisiliquum* had a DMD of less than 35% at 72 h. It is noteworthy that *C. aconitifolius* and *M. oleifera* had a remarkably high DMD, reaching almost 50% and 40% of degradability after 6 h of fermentation, respectively.

### 3.3. Gas production, CH_4_ and CO_2_ Production

The GP results are shown in Table 3. *C. aconitifolius* had the greatest GP (*p* < 0.05) at 6, 12 and 24 h of fermentation, but no differences were observed between *C. aconitifolius* and *C. plectostachyus* at 48 and 72 h. No differences were found between *C. plectostachyus, B. alicastrum* and *M. oleifera* (*p* > 0.05) at 72 h. The lowest GP was recorded with *E. cyclocarpum*, *L. leucocephala and L. latisiliquum*, all of them being different to *C. plectostachyus* at 24, 48 and 72 h (*p* < 0.05).

The kinetics of the GP, CH_4_ and CO_2_ production are shown in Table 4. *C. aconitifolius* had the greatest Vmax (*p* < 0.05), but no differences were observed between *C. aconitifolius* and *C. plectostachyus*. The lowest Vmax was recorded with *E. cyclocarpum* and *L. leucocephala*, both being different to *C. plectostachyus* (*p* < 0.05). The rate was different only with *L. leucocephala* (*p* < 0.05). The greatest Lag was found with *G. sepium*, whereas *A. lebbeck*, *C. aconitifolius*, *E. cyclocarpum* and *L. latisiliquum* recorded the lowest Lag (*p* < 0.05). Regarding the production of CH_4_ and CO_2_, the results varied widely. The highest percentage of CH_4_ was found with *E. cyclocarpum* and *C. plectostachyus*, with more than 50%, whereas in *L. leucocephala* and *G. ulmifolia* it was between 23 and 28%, respectively. *C. plectostachyus* produced the greatest amount of CH_4_ (>23 mL/g DM) at 24 h of fermentation and was statistically different (*p* < 0.05) from the rest of the trees and shrubs, except to *C. aconitifolius* and *M. oleifera*. On the other hand, *B. simaruba*, *E. cyclocarpum*, *G. ulmifolia*, *L. leucocephala*, *L. latisiliquum* and *P. piscipula* decreased the production of CH_4_ (<8 mL/g DM), compared to *C. plectostachyus*, *C. aconitifolius* and *M. oleifera* (*p* < 0.05).

### 3.4. Production of Short-Chain Fatty Acids

Production of SCFA varied between all the treatments (*p* < 0.05) (Table 5), however, no differences were found in the acetic:propionic ratio (*p >* 0.05). It should be noted that *C. aconitifolius*, *M. oleifera and B. alicastrum* produced >70 mmol/L of the total SCFA. Additionally, these treatments had the highest production of acetic, propionic and butyric acids. On the other hand, *G. ulmifolia*, *P. piscipula*, *L. leucocephala*, *B. simaruba*, *E. cyclocarpum* and *L. latisiliquum* exhibited the lowest amount of the total SCFA with values between 35 and 45 mmol/L. *A. lebbeck* produced the greatest amount of branched SCFA, although without differences with *C. plectostachyus* (*p >* 0.05).

### 3.5. Quantification of Ruminal Microbial Populations by Digital Droplet PCR (ddPCR)

Figure 1 displays the copies per μL of the respective target DNA in each sample. When analyzing bacteria, 16S marker (A) showed no significant differences between the foliage, however 23S (B) indicated that bacterial populations varied widely between treatments with higher counts in *B. alicastrum*, *B. simaruba* and *M. oleifera* than in the control and lower bacterial counts in *G. sepium* and *G. ulmifolia*. As expected, the copies per μL of fungi and protozoa were low; nonetheless, the analysis demonstrated that *B. simaruba*, *G. ulmifolia*, *L. latisiliquum* and *L. leucocephala* had higher counts of fungi (C) and *L. latisiliquum*, *L. leucocephala* and *E. cyclocarpum* had higher amounts of protozoa (D) than *C. plectostachyus*. The analysis of archaea by 16S showed a higher quantity of these microorganisms in *E. cyclocarpum* and *B. simaruba*, whereas *G. sepium* had the least amount €. However, when targeting solely *Methanobrevibacter sp.* groups, we found significant differences between *C. aconitifolius*, *G. ulmifolia* and *P. piscipula* with a minimum amount of the RO group (G) than *C. plectostachyus*, whereas the SGMT group were scarcely detected in all samples (H). Besides studying the microbial populations, the abundance of *mcrA* was evaluated and it was found to be reduced only by the foliage of *G. sepium* (F).

### 3.6. Correlation between Condensed Tannins and Saponins, In Vitro Ruminal Fermentation Variables and Microbiota

A correlation analysis between the CT and S content of the foliage, in vitro ruminal fermentation variables and microbial populations is depicted in Figure 2. Most of all the fermentation variables were correlated with each other. A negative correlation was observed between the CT content with CH_4_%, CH_4_ mL, CO_2_ mL and GP_24_ (*p* < 0.05), but no correlation was found with S. Regarding the microorganisms, protozoa were negatively correlated with CH_4_ mL, CO_2_ mL, GP_12–24_ and GP_24_, while archaea had a negative correlation with CO_2_ mL, GP_12–24_ and GP_24_. Interestingly, despite the low amounts of *Methanobrevibacter sp.* In groups RO and SGMT (Figure 1), the analysis displayed a tight positive correlation between these archaea and the foliage (*p* < 0.05). Furthermore, the abundance of protozoa was positively correlated with archaea and fungi, however, this correlation appeared to be specific as both fungi and protozoa were positively related to archaea as a whole, but had a negative correlation with the SGMT group. Finally, fungi were the only group of microorganisms that correlated with CT content.

In order to gain further insight into the relationships between the data, PCA analysis displayed spatially how these variables relate to each other (Figure 3). Interestingly, the foliage of the plants separated into two main groups with further subgroups. On one hand *E. cyclocarpum*, *L. latisiliquum*, *B. simaruba*, *L. leucocephala*, *G. ulmifolia* and *P. piscipula* formed a group that could be split into two subgroups where *L. latisiliquum*, *E. cyclocarpum* and *B. simaruba*, coalesce with archaea, fungi and protozoa, and *L. leucocephala*, *G. ulmifolia* and *P. piscipula* formed a subgroup related to CO_2_%, CT and S. On the other hand, *C. plectostachyus* grouped together with *B. alicastrum* and *M. oleifera*, while *A. lebbeck*, *C. aconitifolius* and *G. sepium* coalesce together, the latter being the outlier of the group. Microorganisms appeared as three main subgroups with the exception of SGMT. Tellingly, CH_4_% and CH_4_ mL were more related to the *C. plectostachyus*, *B. alicastrum* and *M. oleifera* group, whereas CO_2_%, CT and S with *G. ulmifolia* and *P. piscipula*.

## 4. Discussion

### 4.1. Chemical Analysis

The chemical analysis showed that the CP content was between 10 and 25%, thus meeting or exceeding the requirements for cattle [25]. The foliage of *L. leucocephala* and *M. oleifera* had >20% of CP, which are values similar to those reported by Molina et al. [39] and Galindo et al. [19], as well as those from Meale et al. [40] who reported a CP content between 20 and 30% in shrub legumes, and between 7 and 9% in grasses. We used a grass, *C. plectostachyus*, as control whose CP content was 10.26%, which is in line with the reports by Meale et al. [40] and Molina et al. [39], who mention that grasses have a low CP content. An adequate supply of dietary CP, particularly rumen degradable protein, is necessary to increase microbial protein synthesis [41]. Microbial protein supplies one part of the metabolizable protein required by cattle [25].

Ash content was found to be between 4.3 and 11.8% (DM basis) and though the latter is 2.74 times higher than the former, the reported values are within the normal content of foliage of tropical trees and shrubs. As an example, the ash content found in *Azadirachta indica* and *E. cyclocarpum* was 3.9 and 11.8% (DM basis), respectively [19], whereas in another study, the least ash content was reported for *Sapinus saponaria* at 16.2% and the greatest percentage in *Trichantera gigantean* at 21.8% (DM basis) [17]. Thus, our results for *G. sepium*, *G. ulmifolia* and *M. oleifera* are similar with previous studies, however, the amount of ash in *E. cyclocarpum* and *L. leucocephala* was roughly half compared to that reported by others [19,39,40]. Despite the differences in ash content between plants within these studies, none of them relate ash content with the DMD or SCFA [17,19,39,40].

The NDF and ADF content are used as parameters of the quality of a forage. The NDF content refers mainly to the content of hemicellulose, cellulose and lignin; ADF refers to cellulose and lignin, the latter as the main factor affecting the availability of cellulose [42]. The results by Molina et al. [39] agree with the results obtained in this study, since the control treatment (*C. plectostachyus*) presented 78% of NDF. However, the NDF and ADF contents of trees and shrubs analyzed in this study differ from those reported by others [19,40]. These variations may be due to various factors, including environmental conditions such as that temperature that increases the lignification of plants [42].

From the CT and S content results, we found a wide variation among the foliage. *G. ulmifolia* had the greatest content of CT and S (7 and 6.6%, respectively), while *M. oleifera* and *C. plectostachyus* presented the lower content of CT (0.02 and 0.34%, respectively) and S (3.2 and 3.6%, respectively). These values differ from the reports by Delgado et al. [17] who found a high content of CT in the foliage of *L. leucocephala* and *E. cyclocarpum*, and a moderate content of S in the foliage of *A. lebbeck*, *L. leucocephala* and *G. sepium*. The variation between the studies may be a consequence of various factors, among which is the extraction method, the physiological state of the plant, the season of the year in which the sample was collected and the type of soil [43,44].

### 4.2. Dry matter Degradability

In this study, the foliage of *G. ulmifolia*, *P. piscipula*, *L. leucocephala*, *E. cyclocarpum* and *L. latisiliquum* presented less than 35% of DMD. It is important to highlight that almost all these species are leguminous plants, which have a higher lignin content than grasses, thus limiting the DMD [45]. Nonetheless, Molina et al. (39) mention that legumes have a higher degradability than grasses. Since *L. leucocephala* is a tropical species frequently used in ruminant feeding, it was used as reference in this study; however, the DMD of *L. leucocephala* in our study was lower than the values reported by Molina et al. [39], Meale et al. [40] and Gaviria et al. [20]. Other factors, such as the CT content, may also affect the DMD [45].

### 4.3. Gas Production, CH_4_ and CO_2_ Production In Vitro

In this study, a higher GP was observed with *C. aconitifolius* and *C. plectostachyus*, whereas there were no significant differences between *C. plectostachyus* with *M. oleifera*, *B. alicastrum* or *A. lebbeck* at 72 h of incubation. In this regard, Meale et al. [40] reported a higher GP with *G. sepium*, *L leucocephala* and *M. oleifera* (148.5, 126.8 and 187 mL/g DM, respectively) at 24 h of incubation. Likewise, Gaviria et al. [20] reported a GP of 250 mL/g DM at 48 h for *L. leucocephala*. These values are higher than the one found in this study: 82 mL/g MS. Our results can be explained by the low degradability.

As expected, *C. plectostachyus* grass produced more CH_4_ than any of the foliage evaluated. These findings are similar to Galindo et al. [19] who reported a higher CH_4_ production with the grass *Cynodon nlemfuensis* (65.15 mL/g DM) compared to the other foliage. Delgado et al. [17] mention that using the foliage of trees as part of a diet reduces the amount of CH_4_ in ruminants. It should be noted that in addition to having a low GP, *L. leucocephala* produced the least amount of CH_4_, which may be due to its low DMD compared to the other foliage. Additionally, it is important to mention that *C. plectostachyus* probably produced more CH_4_ because it is a Gramineae, since according to the results of Meale et al. [40], grasses produce more CH_4_ than shrub legumes (*L. leucocephala*) and non-shrub legumes (*M. oleifera*). However, in our study, *M. oleifera* had a similar percentage of CH_4_ compared to *C. plectostachyus*.

### 4.4. Production of SCFA

There was a wide variation in the concentration of SCFA and several of the foliage samples had a higher production than the grass *C. plectostachyus*. In this regard, Meale et al. [40] reported differences in the concentration of SCFA from ruminal fermentation between forage species, with *G. sepium* (leguminous shrub), *M. oleifera* (non-leguminous shrub) and *Brachiaria ruziziensis* (grass) being those with the highest amount of SCFA. Some studies report that SCFA concentration could modify CH_4_ production. Moss et al. [46] indicate that acetate and butyrate promote CH_4_ production, while propionate can be considered a competitor of CH_4_ production by regulating H_2_ availability in the rumen. Likewise, Jayanegara et al. [47] mention that the reduction in CH_4_ emissions is related to a decrease in acetate-propionate proportion [9]. In our study, the foliage of *C. aconitifolius*, *M oleifera*, *B. alicastrum* and *A. lebbeck* had the highest concentration of propionic acid (12–14 mmol/L); however, *M. oleifera* and *B. alicastrum* also showed the highest CH_4_ production, though still below the amount of CH_4_ produced by *C. plectostachyus*. In this study, the production of branched SCFA and valeric acid was variable between species, being significantly higher in *A. lebbeck* and *M. oleifera* compared to *G. ulmifolia* and *L. leucocephala*. The concentration of branched-chain SCFA and valeric acid is higher in diets with a high content of rumen-degradable protein, because these SCFA derive from branched-chain amino acids [41]. Furthermore, it has been reported that the ruminal degradability of CP from the leaves of *M. oleifera* is greater than 90% [48], whereas in *L. leucocephala* it is less than 50% [49].

### 4.5. Ruminal Microbial Populations

For DNA quantification, digital droplet PCR (ddPCR) was used because it is a more robust analytical tool for DNA quantification due to its higher sensitivity, lower sample inhibitory effects, as well as allowing an absolute quantification without the need for a known standard [50,51,52]. Additionally, ddPCR has been proven to be accurate when assessing complex and environmental microbial samples [53,54,55,56], such as the ruminal microbiota.

The four main groups of microorganisms in the ruminal microbiota, as well as specific markers for archaea, were quantified by ddPCR. The analysis of bacteria with a 16S marker showed no significant differences between all of the tested foliage, though some differences were found with 23S (Figure 1). Perhaps none of the treatments affect bacterial populations, however, even if a foliage reduced certain genera, the results could be masked because a 16S copy number in bacteria varies from 2 to 10 copies per genome depending on the clade and genera. Nonetheless, ddPCR has been shown to accurately determine 16S copy number and even though DNA quality can cause an underestimation, the relative abundance of bacteria is not affected [56]. Furthermore, as ruminal microbiota sequences increase through multiple sequencing efforts, any technology based on targeted amplification has limitations as it is prone to primer mismatch with the targeted sequences causing a bias [57,58]. Notwithstanding, our ddPCR results are similar to other works using different techniques. For example, Anantasook et al. [59] and Vasta et al. [60] did not find changes in bacterial counts with q-PCR using a legume tree containing CT and S or supplemented a diet with 6.4% tannins, respectively. Similarly, Wallace et al. [61] using metaprofiling, i.e., 16S targeted sequencing, found the same counts for bacteria between high and low methane-emitting cattle. Similarly, Thomas et al. [58] did not find changes at the phylum level in bacteria through a metagenomic approach. These results indicate that ruminal microbiota displays both redundancy and resiliency [62].

Concerning the archaea, this group was evaluated with a 16S and *mcrA*, as well as a pair of specific primers for *Methanobrevibacter*. Surprisingly, our results using 16S and *mcrA* differed, although the latter has been established as a reliable marker to monitor methanogenic archaea [11]. This may be partially explained by the different number of 16S copies per genome between crenarchaea and euryarchaea, the former with 1 copy/genome while the latter has an average of 1.9 copies [50]. Moreover, recently it has been shown that the *mcrA* gene is more ubiquitous and has other enzymatic functions than previously reported [63]. Our results with 16S RO and 16S SGMT, with the exception of the foliage of *E. cyclocarpum*, did not coincide with the 16S and *mcrA*, suggesting either a mismatch during amplification with the *mcrA* primer [63] or a specific enrichment of certain archaeal taxa with each foliage. Nonetheless, the lower counts of *mcrA*, RO and SGMT were found in *C. aconitifolius*, the foliage with the highest concentration of propionic acid (Table 5, and *G. ulmifolia,* the foliage with the lowest CH_4_ production (Table 4) and highest CT content (Table 1).

The counts of fungi and protozoa were low, as expected, as they are represented by a low number of cells. Interestingly, the foliage with the higher quantity of fungi and protozoa was also the one containing a higher amount of CT (*G. ulmifolia*). These results are similar to Vasta et al. [60] as a diet with 6.4% of tannins increased the number of protozoa, however, this is in contrast with the findings by Anantasook et al. [59] as a legume tree containing CT and S diminished both fungi and protozoa.

### 4.6. Correlation between Foliage, Methane Production, Fermentation Variables and Ruminal Microbial Populations

A correlation analysis of our data (Figure 2) showed interesting results between the fermentation metrics, CT and microorganisms. As previously reported, the CT was negatively correlated with CH_4_, however, protozoa were also inversely correlated with CH_4_ despite being the target of many strategies for diminishing CH_4_ emissions. Furthermore, protozoa appeared positively correlated with archaea and fungi, as could be expected, as some of the former share a symbiotic relationship with protozoa. Interestingly, despite the positive relationship between protozoa and archaea, the opposite relationship was found with the SGMT group. This observation is similar to those from Li et al. [64] and Danielsson et al. [65] who respectively found that the *Methanobrevibacter* RO clade was negatively related to SGMT, and that the RO group was associated with low CH_4_ emitters, whereas SGMT was related with a higher CH_4_ production. Finally, both RO and SGMT groups appeared associated with the type of foliage, thus suggesting that the composition of the foliage has an impact on these archaea. The PCA analysis also displayed this antagonistic relationship between *Methanobrevibacter* RO and SGMT clades. Moreover, the PCA grouped together most of the plants that poses foliage with high and intermediate quantities of CT (Figure 3).

The use of trees and shrubs as a means to diminish ruminal CH_4_ production is not straightforward. As our present work demonstrates, the effect of foliage containing different types and quantities of secondary metabolites affect fermentation metrics and ruminal microbiota; however, the correlation between all of these variables is complex. The decrease in CH_4_ related to the CT concentration in the foliage can be partially attributed to a decrease in the DMD of the nutrients, which is a result of complexes formed by tannins with proteins and carbohydrates [57]. Any reduction in fiber degradation is likely to reduce CH_4_ formation [59]. It is important to note that both condensed and hydrolysable tannins lower ruminal CH_4_ emissions. However, hydrolysable tannins have the capacity to reduce CH_4_ production without affecting degradability, probably due to gallic acid, a subunit of this type of tannin. In contrast, the decrease in CH_4_ emissions from CT is attributed to a lower ruminal degradability or decreased DM intake. Additionally, plants from tropical regions have higher concentrations of CT (0.7 to 23.8 g/100 g MS) than those from temperate zones (0.04 to 9.9 g/100 g DM) [66].

As would be expected, the higher DMD observed in *B. alicastrum*, *C. aconitifolius*, *C. plectostachyus* and *M. oleifera* (Table 2) was concomitant with a greater SCFA production (Table 5), whereas the foliage with the lower DMD was mostly the same with the least quantity of SCFA, including *G. ulmifolia*, *E. cyclocarpum*, *L. latisiliquum*, *P. piscipula* and *L. leucocephala*. Tellingly, this distribution became evident with the PCA analysis (Figure 3) with the former group being more correlated with CH_4_ production and 16S SGMT, and the latter with the secondary metabolites 16S archaea, 18S fungi and 18S protozoa.

The effect of the foliage of trees and shrubs in ruminal microbiota could be masked by the resiliency, redundancy and host specificity of the ruminal microorganisms [62]. Several authors have found that either adding tannins, antibiotics or diets do not affect microorganisms counts or only effect them to some extent [58,59,60,61,64,67,68]. As recently described by Martínez-Alvaro et al. [69], the rumen microbiome consists of 10 main functional niches, being the Methanobacteriales key to methanogenesis. However, the variables that explained CH_4_ production were pathways unrelated to the former, thus indicating that the microbial consortia as a whole is responsible for CH_4_ production and not specific to archaea.

## 5. Conclusions

Under the conditions of this study, we conclude that the foliage of tropical trees and shrubs evaluated here showed variability in their chemical composition; they have a high nutritional value and the potential to decrease CH_4_ production compared to *C. plectostachyus*. The CH_4_ reduction is probably due to the CT content in the foliage. It is essential to note that in this study, the foliage was evaluated individually, thus further research where the foliage is evaluated as part of a diet is pending. We consider that it is sensible to include the foliage of tropical plants in the diet, as a clean and agro-ecological feeding strategy for ruminants; however, further in vitro and in vivo studies remain to be conducted. Nonetheless, based upon the nutritional value and the lower CH_4_ emissions, we suggest to use the foliage of *E. cyclocarpum*, *G. ulmifolia*, *L. latisiliquum*, *L. leucocephala* and *P. piscipula* in the diets of ruminants.

Interestingly, our evaluation of ruminal microorganisms is more in line with other molecular and genomic findings than with the previous status quo, which mentions that mainly archaea and protozoa are responsible of CH_4_ production. Contrary to this idea, the sole abundance of these microorganism do not enhance CH_4_ production. Our analysis showed that the abundance of ruminal microorganisms was not tightly related with CH_4_ production and other fermentation variables. Perhaps further molecular and genomic approaches may help elucidate the complex ruminal microbial ecosystem regarding CH_4_ emissions.

## Figures and Tables

**Figure 1 animals-12-02628-f001:**
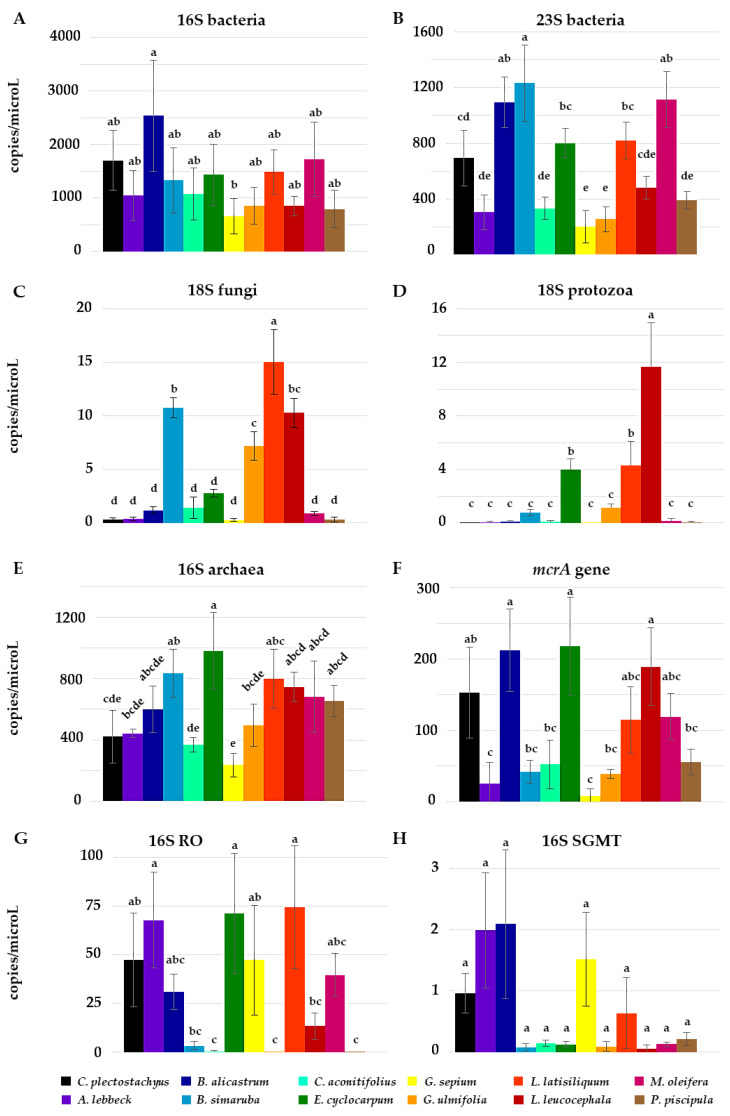
Foliage of tropical trees and shrubs modify the ruminal microbiota during in vitro fermentation. The four main ruminal microbial populations were quantified with primers for 16S and 23S of bacteria ((**A**) and (**B**), respectively), 18S of fungi (**C**), 18S of protozoa (**D**) and 16S of archaea (**E**). Archaea subgroups were evaluated with the *methyl co-enzyme M reductase alpha subunit* (*mcrA*) gene (**F**) and with clade specific primers for *Methanobrevibacter* sp. subgroups RO (*M. ruminantium*, *M. olleyae*) and SGMT (*M. smithii*, *M. gottschalkii*, *M. millerae*, *M. thaueri*) ((**G**) and (**H**), respectively). Color bars represent each of the tree and shrub species. ^a–e^ Different letters in the same column denote differences (*p* < 0.05).

**Figure 2 animals-12-02628-f002:**
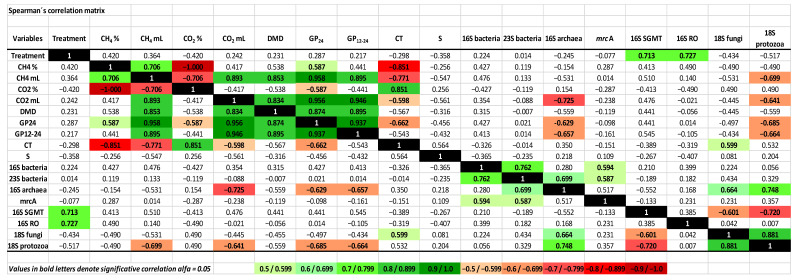
Heatmap of the correlation between methane and carbon dioxide production with fermentation variables, secondary metabolites and microbial populations. Values with a *p* < 0.05 are displayed in bold letters. Shades of green or red color, where darker color denotes higher significance values, indicate positive and negative correlation, respectively. Uncolored values indicate *p* > 0.05. Black cells indicate correlation between the same variable. CH_4_% and CO_2_%: percentage of CH_4_ or CO_2_ in GP_12–24_; CH_4_ mL and CO_2_ mL: CH_4_ or CO_2_ in GP_12–24_. DMD: dry matter degradability at 24 h (mg/g DM); GP_12–24_: gas produced between 12 and 24 h of fermentation (mL/g DM); GP_24_: accumulated gas production at 24 h (mL/g DM); CT: condensed tannins (%); S: saponins (%); 16S bacteria, 23S bacteria, 16S archaea, *mcrA* (*methyl co-enzyme M reductase alpha subunit*), 16S SGMT (*M. smithii*, *M. gottschalkii*, *M. millerae* and *M. thaueri*), 16S RO (*M. ruminantium* and *M. olleyae*), 18S fungi, 18S protozoa: copies/µL.; 16S bacteria, 23S bacteria, 16S archaea, *mcrA* (*methyl co-enzyme M reductase alpha subunit*), 16S SGMT (*M. smithii*, *M. gottschalkii*, *M. millerae* and *M. thaueri*), 16SRO (*M. ruminantium* and *M. olleyae*), 18S fungi, 18S protozoa: copies/µL.

**Figure 3 animals-12-02628-f003:**
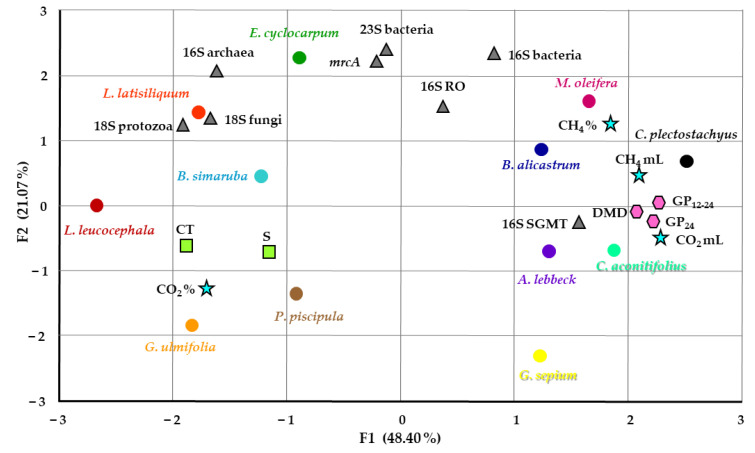
Principal Component Analysis (PCA) of foliage from twelve plants and their relationship with methane and carbon dioxide production, fermentation variables, content of secondary metabolites and microbial populations. PCA analysis spatially resolved differences between the foliage where two factors account for 69% of the differences between all variables. Circles denote the foliage in their respective colors code of Figure 1, blue stars show CH_4_ and CO_2_, pink hexagons indicate fermentation variables, lime green squares represent secondary metabolites and grey triangles designate the corresponding microbial populations. CH_4_% and CO_2_%: percentage of CH_4_ or CO_2_ in GP_12–__24_; CH_4_ mL and CO_2_ mL: CH_4_ or CO_2_ in GP_12–__24_. DMD: dry matter degradability at 24 h (mg/g DM); GP_12–24_: gas produced between 12 and 24 h of fermentation (mL/g DM); GP_24_: accumulated gas production at 24 h (mL/g DM); CT: condensed tannins (%); S: saponins (%); 16S bacteria, 23S bacteria, 16S archaea, *mcrA* (*methyl co-enzyme M reductase alpha subunit*), 16S SGMT (*M. smithii*, *M. gottschalkii*, *M. millerae* and *M. thaueri*), 16S RO (*M. ruminantium* and *M. olleyae*), 18S fungi, 18S protozoa: copies/µL.

**Table 1 animals-12-02628-t001:** Chemical composition of the foliage of tropical trees and shrubs.

Species	Chemical Composition, (% DM)							
	OM	CP	Ash	NDF	ADF	EE	CT	S
*C. plectostachyus*	92.7	10.3	7.3	78.3	39.4	2.1	0.34	3.6
*A. lebbeck*	91.9	16.9	8.1	43.7	27.9	5.0	1.07	3.8
*B. alicastrum*	90.2	16.6	9.8	45.1	35.5	2.4	2.18	4.5
*B. simaruba*	88.2	11.6	11.8	45.2	44.1	3.3	4.62	3.9
*C. aconitifolius*	92.5	19.2	7.5	21.7	18.1	5.7	1.03	3.6
*E. cyclocarpum*	93.5	14.2	6.5	67.8	45.3	2.6	1.27	5.0
*G. sepium*	89.1	16.6	10.9	51.5	36.9	3.9	1.55	3.8
*G. ulmifolia*	89.7	16.4	10.3	55.4	37.9	5.2	7.00	6.6
*L. leucocephala*	95.7	22.9	4.3	54.2	32.1	2.7	4.19	4.3
*L. latisiliquum*	94.1	14.1	5.9	54.5	27.8	3.5	3.75	3.3
*M. oleifera*	91.3	25.4	8.7	16.2	10.5	4.8	0.02	3.2
*P. piscipula*	90.4	14.3	9.6	58.9	43.3	3.5	1.55	4.6

DM: dry matter; OM: organic matter; CP: crude protein; NDF: neutral detergent fiber; ADF: acid detergent fiber; EE: ether extract; CT: condensed tannins; and S: saponins.

**Table 2 animals-12-02628-t002:** Dry matter degradability (DMD) of the foliage of tropical trees and shrubs at different in vitro ruminal fermentation times (mg/g DM).

Species	Hours				
	6	12	24	48	72
*C. plectostachyus*	233.81 ^cd^	301.84 ^cd^	322.16 ^bc^	518.77 ^cd^	578.52 ^bc^
*A. lebbeck*	268.64 ^bc^	280.61 ^cde^	333.47 ^bc^	433.65 ^de^	514.91 ^cd^
*B. alicastrum*	317.54 ^b^	462.02 ^b^	431.90 ^b^	679.64 ^a^	698.19 ^a^
*B. simaruba*	167.99 ^de^	217.35 ^def^	259.94 ^cd^	326.50 ^fg^	386.15 ^e^
*C. aconitifolius*	464.00 ^a^	565.00 ^a^	582.76 ^a^	615.82 ^ab^	619.95 ^b^
*E. cyclocarpum*	133.81 ^e^	172.29 ^f^	199.31 ^d^	240.72 ^hg^	287.47 ^f^
*G. sepium*	295.03 ^bc^	362.97 ^bc^	388.07 ^b^	402.57 ^ef^	467.19 ^d^
*G. ulmifolia*	105.28 ^e^	184.02 ^ef^	196.68 ^d^	232.49 ^h^	277.41 ^f^
*L. leucocephala*	131.91 ^e^	168.64 ^f^	194.31 ^d^	265.42 ^gh^	339.48 ^ef^
*L. latisiliquum*	121.07 ^e^	131.45 ^f^	161.29 ^d^	288.54 ^gh^	310.29 ^f^
*M. oleifera*	394.77 ^a^	411.87 ^b^	430.04 ^b^	550.15 ^bc^	574.13 ^bc^
*P. piscipula*	151.10 ^e^	188.45 ^ef^	194.91 ^d^	282.34 ^gh^	313.30 ^f^
SEM	13.35	16.32	17.08	17.67	15.58

^a–h^ Different letters in the same column denote differences (*p* < 0.05). SEM = standard error mean.

**Table 3 animals-12-02628-t003:** Gas production (GP) of the foliage of tropical trees and shrubs at different in vitro ruminal fermentation times (mL/g DM).

Species	Hours				
	6	12	24	48	72
*C. plectostachyus*	57.63 ^bcde^	95.66 ^bcd^	142.01 ^b^	208.04 ^ab^	236.28 ^ab^
*A. lebbeck*	65.79 ^bc^	93.24 ^bcd^	119.12 ^bc^	143.46 ^cd^	151.82 ^c^
*B. alicastrum*	66.80 ^b^	113.73 ^b^	146.15 ^b^	183.04 ^bc^	207.30 ^b^
*B. simaruba*	34.68 ^e^	57.51 ^de^	76.78 ^d^	106.39 ^def^	132.16 ^c^
*C. aconitifolius*	103.78 ^a^	163.07 ^a^	208.10 ^a^	241.08 ^a^	258.70 ^a^
*E. cyclocarpum*	35.36 ^e^	52.49 ^e^	65.71 ^d^	77.24 ^f^	85.42 ^de^
*G. sepium*	42.32 ^cde^	62.42 ^cde^	94.38 ^cd^	125.40 ^de^	135.16 ^c^
*G. ulmifolia*	35.07 ^e^	53.09 ^e^	69.97 ^d^	101.25 ^ef^	115.58 ^cd^
*L. leucocephala*	37.19 ^de^	59.73 ^cde^	61.74 ^d^	64.70 ^f^	67.88 ^e^
*L. latisiliquum*	40.18 ^de^	56.80 ^de^	68.56 ^d^	80.62 ^f^	86.37 ^de^
*M. oleifera*	59.44 ^bcd^	96.78 ^bc^	134.06 ^b^	188.59 ^b^	209.08 ^b^
*P. piscipula*	38.15 ^de^	57.27 ^de^	69.81 ^d^	91.89 ^ef^	113.51 ^cd^
SEM	2.44	4.12	4.98	6.00	6.51

^a–f^ Different letters in the same column denote differences (*p* < 0.05). SEM = standard error mean.

**Table 4 animals-12-02628-t004:** Kinetics parameters from fitted curves and production of methane (CH_4_) and carbon dioxide (CO_2_) of the foliage of tropical trees and shrubs in ruminal fermentation in vitro.

Species	Vmax	Rate	Lag	CH_4_%	CO_2_%	CH_4_ mL	CO_2_ mL
*C. plectostachyus*	214.90 ^ab^	0.034 ^b^	3.411 ^ab^	50.3 ^a^	49.7 ^c^	23.31 ^a^	23.03 ^ab^
*A. lebbeck*	152.09 ^cde^	0.060 ^b^	1.105 ^b^	48.8 ^ab^	51.2 ^bc^	12.63 ^bcd^	13.25 ^bc^
*B. alicastrum*	171.00 ^bcd^	0.051 ^b^	1.501 ^ab^	43.4 ^abc^	56.6 ^abc^	14.07 ^bc^	18.35 ^abc^
*B. simaruba*	121.96 ^cdef^	0.048 ^b^	1.312 ^ab^	38.2 ^abc^	61.8 ^abc^	7.36 ^cde^	11.91 ^cd^
*C. aconitifolius*	237.81 ^a^	0.063 ^b^	0.475 ^b^	44.3 ^abc^	55.7 ^abc^	19.45 ^ab^	25.07 ^a^
*E. cyclocarpum*	79.56 ^fg^	0.059 ^b^	0.732 ^b^	51.8 ^a^	48.2 ^c^	6.85 ^cde^	6.37 ^de^
*G. sepium*	128.72 ^cdef^	0.043 ^b^	5.394 ^a^	42.6 ^abc^	57.4 ^abc^	13.61 ^bc^	18.35 ^abc^
*G. ulmifolia*	112.10 ^efg^	0.078 ^b^	1.918 ^ab^	23.7 ^c^	76.3 ^a^	4.00 ^e^	12.88 ^bcd^
*L. leucocephala*	57.19 ^g^	0.246 ^a^	2.588 ^ab^	28.0 ^bc^	72.0 ^ab^	0.56 ^e^	1.44 ^e^
*L. latisiliquum*	94.80 ^efg^	0.062 ^b^	0.737 ^b^	35.6 ^abc^	64.4 ^abc^	4.18 ^e^	7.57 ^de^
*M. oleifera*	175.33 ^bc^	0.034 ^b^	4.255 ^ab^	46.8 ^ab^	53.7 ^bc^	17.44 ^ab^	20.02 ^abc^
*P. piscipula*	112.42 ^defg^	0.055 ^b^	2.826 ^ab^	41.5 ^abc^	58.5 ^abc^	5.20 ^de^	7.33 ^de^
SEM	6.037	0.008	0.280	1.71	1.71	0.97	1.16

Vmax: volume maximum of GP, mL/g DM; Lag: lag phase, h^−1^; rate: rate of gas production/h; CH_4_% and CO_2_%: percentage of CH_4_ or CO_2_ in GP_12–24_; CH_4_ mL and CO_2_ mL: CH_4_ or CO_2_ in GP_12–__24_. ^a–g^ Different letters in the same column denote differences (*p* < 0.05). SEM = standard error mean.

**Table 5 animals-12-02628-t005:** Production of short-chain fatty acids (SCFA) at 72 h of in vitro ruminal fermentation of the foliage of tropical trees and shrubs.

Species	SCFA (mmol/L)							
	Acetic	Propionic	Isobutyric	Butyric	Isovaleric	Valeric	Total	Acetic: Propionic Ratio
*C. plectostachyus*	40.64 ^bc^	10.45 ^bc^	0.87 ^ab^	5.71 ^abc^	1.33 ^abc^	0.91 ^ab^	59.90 ^bc^	3.89
*A lebbeck*	42.34 ^abc^	12.21 ^ab^	1.09 ^a^	5.66 ^abc^	1.95 ^a^	1.02 ^a^	64.27 ^ab^	3.47
*B. alicastrum*	48.84 ^ab^	12.16 ^ab^	0.92 ^ab^	7.86 ^a^	1.53 ^abc^	0.97 ^a^	72.28 ^ab^	4.02
*B. simaruba*	28.97 ^d^	7.23 ^cd^	0.53 ^ab^	3.31 ^d^	0.84 ^bc^	0.49 ^bc^	41.38 ^d^	4.01
*C. aconitifolius*	50.07 ^ab^	14.05 ^a^	0.90 ^ab^	7.68 ^a^	1.48 ^abc^	1.05 ^a^	75.23	3.69
*E. cyclocarpum*	29.20 ^d^	7.77 ^cd^	0.61 ^ab^	3.75 ^cd^	0.97 ^bc^	0.45 ^c^	42.75 ^d^	3.76
*G. sepium*	34.19 ^cd^	8.11 ^cd^	0.66 ^ab^	3.52 ^cd^	1.21 ^abc^	0.80 ^abc^	48.49 ^cd^	4.22
*G. ulmifolia*	24.67 ^d^	6.08 ^d^	0.34 ^b^	3.54 ^cd^	0.58 ^c^	0.45 ^c^	35.67 ^d^	4.06
*L. latisiliquum*	31.40 ^cd^	7.82 ^cd^	0.64 ^ab^	4.55 ^bcd^	1.00 ^abc^	0.52 ^bc^	45.94 ^cd^	4.01
*L. leucocephala*	29.57 ^d^	7.51 ^cd^	0.43 ^ab^	3.51 ^cd^	0.66 ^c^	0.46 ^c^	42.14 ^d^	3.94
*M. oleifera*	52.70 ^a^	13.19 ^ab^	0.95 ^ab^	6.62 ^ab^	1.65 ^ab^	1.15 ^a^	76.26 ^a^	3.99
*P. piscipula*	29.07 ^d^	7.28 ^cd^	0.55 ^ab^	3.51 ^cd^	0.88 ^bc^	0.47 ^c^	41.77 ^d^	3.99
SEM	1.46	0.42	0.047	0.27	0.077	0.044	2.19	0.067

^a–d^ Different letters in the same column denote differences (*p* < 0.05). SEM = standard error mean.

## Data Availability

The data presented in this study are available on request from the corresponding authors.

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
