# Peer review of "Foliage of Tropical Trees and Shrubs and Their Secondary Metabolites Modify In Vitro Ruminal Fermentation, Methane and Gas Production without a Tight Correlation with the Microbiota"

_animals, 2022, doi:10.3390/ani12192628_

Round 1
Reviewer 1 Report
The manuscript entitled ‘Foliage of tropical trees and shrubs and their secondary metabolites modify in vitro ruminal fermentation, methane and gas production without a tight correlation with the microbiota’ is well-organized and prepared paper presenting the potential of different plants to diminish methane production originated from ruminal fermentation. This topis is very important and closely related to our environment.
I really appreciated Authors for using a lot of method in this manuscript (especially such related with rumen microbiome).
The language of the manuscript is good and allows readers to understand its content and message.
I have only some remarks and suggestions, which should be done before the manuscript could be accepted for publication in Animals.
I suggest Authors to use short-chain fatty acids (SCFA) instead of volatile fatty acids (VFA) throughout the whole manuscript.
Simple summary
L20: should be ‘these’;
2.5. Determination of CH4 and CO2
What was the temperature programme of the used method? There is also a lack of reference for estimating methane and carbon dioxide. Please include it to the paper.
2.6. Production of volatile fatty acids
There is a lack of describing methodology and the temperature programme in detail. Please include it.
3. Results
L285-287: The results obtained for B. simaruba (7.36 Ml/g DM) also fit this description. Please check it.
L308-319: This section should be placed in the Materials and Methods. Please improve it.
L329: should be ‘the’;
Figure 1 and Figure 2
It is suggested to shorten this description. Part of the information was in the Materials and Methods section.
5. Conclusions
Please provide information on which plants could be included in the diet for ruminants.
Reviewer 2 Report
The manuscript examined the effects of Foliage of tropical trees and shrubs and their secondary metabolite on in vitro ruminal fermentation, methane and gas pro- 3 duction without a tight correlation with the microbiota. Generally, the study was well conceived, conducted, and presented. The manuscript contains useful information with room for ruminant nutrition. Thus, I recommend the acceptance of the manuscript for publication subject to a minor revision.
Line 37: Replace ‘oscillated’ with ‘ranged’
Line 41: Replace ‘We’ with ‘It was’
Line 44-45: Only words in the abstract but not in the title should be listed as keywords to enhance the searchability of the manuscript when published.
Line 220: Were the PCR/microbiota data normally distributed? If no, what step did you take to ensure the data are normally distributed before subjecting to analysis of variance.
Table 2: Provide the p value for each parameter
Table 3; Provide the p value for each parameter
Table 4; Provide the p value for each parameter
Table 5; Provide the p value for each parameter.
Line 511: Delete ‘Quantificat
ions of ‘
Reviewer 3 Report
The manuscript deals with a very interesting topic, especially in today's time of global warming and the search for solutions by the research community.
The research was very extensive with a large number of different plant species and analyzed parameters, all of which can be seen in the results of the research.
However, I miss a little reflection on the results in the Discussion. The discussion repeats part of the results only with a comparison with the researches listed in the References. The real Discussion follows only in the part of correlations, which represents the most important part of the entire Manuscript.
Furthermore, I agree with everything in your Conclusion, except perhaps the last sentence, because I do not see on what basis you came to such a conclusion.
Reviewer 4 Report
I congratulate the authors on this extensive screening work on the potential of CT and saponins contained in the leaves of tropical vegetation to limit the in vitro production of CH4 and CO2 from OM fermentation.
I would like to give a suggestion for future research: as a control, the grass Cynodon plectostachyus was used, which has a very different NDF and protein contents than the other substrates, and thus a different OM composition.
I wonder if the timescale of dry matter degradability can be very different between this and other studied biomasses. This is why I would have found it more correct to also do a degradability assessment at 240 h, the time at which even for fibre-rich feed the degradability of NDF is complete. This ensures that all OM of all biomass is degraded and the gas production obtained is as close as possible to the potential for that biomass.
